Mutational spectra of SARS-CoV-2 isolated from animals

Elaswad Ahmed 1 ahmed_elaswad@vet.suez.edu.eg
http://orcid.org/0000-0003-1017-6280 Fawzy Mohamed 2
Basiouni Shereen 3
Shehata Awad A. 4 5 awad.shehata@vet.usc.edu.eg
1 Department of Animal Wealth Development, Faculty of Veterinary Medicine, Suez Canal University , Ismailia , Egypt
2 Department of Virology, Faculty of Veterinary Medicine, Suez Canal University , Ismailia , Egypt
3 Clinical Pathology Department, Faculty of Veterinary Medicine, Benha University , Benha , Egypt
4 Avian and Rabbit Diseases Department, Faculty of Veterinary Medicine, Sadat City University , Sadat City , Egypt
5 Research and Development Section, PerNaturam GmbH , Gödenroth , Germany
Gillespie Joseph
Electronic publication date: 2020 Dec 18
Publication date: 2020
Volume: 8
Electronic Location ID: e10609
Received 2020 Jul 31; Accepted 2020 Nov 29
Copyright: © 2020 Elaswad et al.
Copyright year: 2020
Copyright holder: Elaswad et al.
License: This is an open access article distributed under the terms of the Creative Commons Attribution License, which permits unrestricted use, distribution, reproduction and adaptation in any medium and for any purpose provided that it is properly attributed. For attribution, the original author(s), title, publication source (PeerJ) and either DOI or URL of the article must be cited.
License URL: https://creativecommons.org/licenses/by/4.0/

Keywords: Alignment, Animals, Evolution, Phylogenetic analysis, Mink, SARS-CoV-2, Sequencing

Funding: The authors received no funding for this work.

==============================
Coronaviruses are ubiquitous and infect a wide spectrum of animals and humans. The newly emerged severe acute respiratory syndrome coronavirus-2 (SARS-CoV-2) has become a worldwide pandemic. To address the role that animals may play in the evolution of SARS-CoV-2, the full genome sequences of SARS-CoV-2 isolated from animals were compared with SARS-CoV-2 human isolates from the same clade and geographic region. Phylogenetic analysis of SARS-CoV-2 isolated from the cat, dog, mink, mouse, and tiger revealed a close relationship with SARS-CoV-2 human isolates from the same clade and geographic region with sequence identities of 99.94–99.99%. The deduced amino acid sequence of spike (S) protein revealed the presence of a furin cleavage site (682RRAR▾685), which did not differ among all SARS-CoV-2 isolates from animals and humans. SARS-CoV-2 isolates from minks exhibited two amino acid substitutions (G261D, A262S) in the N-terminal domain of S protein and four (L452M, Y453F, F486L, N501T) in the receptor-binding motif (RBM). In the mouse, the S protein had two amino acid substitutions, one in the RBM (Q498H) and the other (N969S) in the heptad repeat 1. SARS-CoV-2 isolated from minks furtherly exhibited three unique amino acid substitutions in the nucleocapsid (N)protein. In the cat, two unique amino acid substitutions were discovered in the N (T247I) and matrix (T175M) proteins. Additionally, SARS-CoV-2 isolated from minks possessed sixteen, four, and two unique amino acid substitutions in the open reading frame 1ab (ORF1ab), ORF3a, and ORF6, respectively. Dog and cat SARS-CoV-2 isolates showed one and seven unique amino acid substitutions in ORF1ab, respectively. Further studies may be necessary to determine the pathogenic significance of these amino acid substitutions to understand the molecular epidemiology and evolution of SARS-CoV-2.

Introduction

The outbreak of the novel coronavirus disease 2019 (COVID-19) caused by the severe acute respiratory syndrome coronavirus-2 (SARS-CoV-2) has been reported in Wuhan City, Hubei Province, China, and posed unprecedented challenges to global health. Despite the ongoing efforts to control the COVID-19 outbreak, the disease is still spreading worldwide with increasing numbers of infected cases and deaths. The World Health Organization (WHO) officially declared it as a pandemic on 11 March 2020 (Wu et al., 2020; Zhou et al., 2020).

Severe acute respiratory syndrome coronavirus-2 belongs to the order Nidovirales, suborder Cornidovirineae, family Coronaviridae, subfamily Orthocoronavirinae, genus Betacoronavirus, and subgenus Sarbecovirus. The virus is enveloped with a single-stranded positive-sense RNA genome of 29,903 nucleotides. SARS-CoV-2 has a monopartite genome that consists of two untranslated regions (5′ and 3′ UTRs) and 11 open reading frames (ORFs) encoding 27 proteins. ORF1ab constitutes the first two-thirds of the genome (21,290 nucleotides) and encodes 16 non-structural proteins (nsp1–nsp16). The last third of the genome encodes four structural and six accessory proteins. The structural proteins are the Spike (S), Envelope (E), Matrix (M), and Nucleocapsid (N) proteins, while the accessory proteins include the ORF3a, ORF6, ORF7a, ORF7b, ORF8, and ORF10 (Chan et al., 2020).

The N protein coils the RNA segment into a helical nucleocapsid that is incorporated into a virion of icosahedral symmetry, whereas the M and E proteins are required for virus morphogenesis, assembly, and budding (Paraskevis et al., 2020). The S glycoprotein (1273 aa) is a fusion viral protein that consists of two subunits: S1 (681 aa) and S2 (588 aa). The S1 subunit comprises a signal peptide (SP), an N-terminal domain (NTD), and a receptor-binding domain (RBD) that interacts with the angiotensin-converting enzyme 2 (ACE2) (Hoffmann et al., 2020; Letko, Marzi & Munster, 2020; Walls et al., 2020; Zhao et al., 2020). The S2 subunit comprises a fusion protein (FP) that mediates membrane fusion and two heptad repeats known as HR1 and HR2, which form the coiled structures surrounded by the protein ectodomain. A furin cleavage site (RRAR▾) for furin proteases exists between the interface of S1 and S2 subunits of SARS-CoV-2 but not SARS-CoV (Canrong et al., 2020; Coutard et al., 2020). It mediates the initial attachment of the virus to the host-cell receptors and subsequent fusion with the cell membrane allowing virus entry. Like SARS-CoV, SARS-CoV-2 binds to ACE2 for cell entry as mediated by the viral surface S glycoprotein (Hoffmann et al., 2020; Walls et al., 2020; Yan et al., 2020; Zhou et al., 2020). Subsequently, the S protein is cleaved by the transmembrane serine protease 2 (TMPRSS2) to S1 and S2 subunits (Hoffmann et al., 2020) allowing the entry of the virion core.

There is a broad acceptance that SARS-CoV-2 has an animal origin, although the animal reservoir has not yet been identified (Abdel-Moneim & Abdelwhab, 2020). Bats may be considered a potential reservoir host for SARS-CoV-2 since they are the reservoir of several SARS related coronaviruses. One of these viruses is the horseshoe bat (Rhinolophus affinis) coronavirus RaTG13, which shares a high identity (96.3%) with SARS-CoV-2 at the genome level (Helmy et al., 2020). Wong et al. (2020) suggested that pangolin might be the intermediate host as it shares 98% identity with SARS-CoV-2 at the receptor-binding motif (RBM) of S protein. The receptor recognition by the S glycoprotein is the major determinant of host range, tissue tropism, and pathogenesis of coronaviruses. Human-to-human transmission has been confirmed even from asymptomatic carriers and pre-symptomatic infected persons (Rothe et al., 2020; Zou et al., 2020). Several cases of SARS-CoV-2 infections in animal hosts (cat, dog, mink, and tiger) have been reported (Gollakner & Capua, 2020; Tazerji et al., 2020), although none of them developed typical clinical features as COVID-19 patients. The dynamics of the disease concerning the transmission of the virus from humans to animals and vice versa requires further explanation. Additionally, there is inadequate information on whether animals have a role in the evolution of SARS-CoV-2. Hence, the mutation rate could drive viral evolution and genome variability, thereby enabling viruses to escape host immunity and develop drug resistance. In the present study, the genetic diversity and mutations in SARS-CoV-2 isolated from cats, dogs, mouse, minks, and tiger were characterized and compared with human isolates to investigate the mutational spectra that might play a role in the virus evolution, transmission, and pathogenesis.

Materials and Methods

SARS-CoV-2 sequences used for genetic analysis

A total of 310 complete coronavirus genomes were used in the current study. All genomes were downloaded from the Global Initiative on Sharing All Influenza Data (GISAID) and the National Center for Biotechnology Information (NCBI) databases. These genomes included: 157 genomes for SARS-CoV-2 isolates from animals (cat (n = 6), dog (n = 2), mink (n = 147), mouse (n = 1), and tiger (n = 1)), 150 human SARS-CoV-2 isolates from Belgium, China, England, France, Hong Kong, Netherlands, Spain, and the USA, SARS-CoV-2 reference genome (Wuhan-Hu-1), pangolin coronavirus (MT121216.1), and bat coronavirus RaTG13 (MN996532.2). The genomes of human isolates were selected based on three conditions: (1) They belonged to the same clade (GISAID clades) as the animal isolates, (2) They originated from the same place (country), and (3) They were collected for sequencing at the same time as animal isolates whenever possible. For example, the SARS-CoV-2 isolate from the tiger was collected in April 2020 from New York, USA. Therefore, we selected 30 human isolates from New York collected in the same month and from the same clade (GH). For each animal species, 30 human SARS-CoV-2 genomes were selected for comparison. Further data about the 310 genomes used in the current study including the virus isolate, accession number, collection date, clade, genome length, host, geographic area, and percentage of ambiguous bases (%N) are listed in Table S1.

Genetic analysis

Identification of mutations

Only the coding gene sequences were used for the analysis. Sequences were aligned using Kalign3 (Lassmann, 2020), and the single nucleotide polymorphisms were analyzed using the SNiPlay pipeline (Dereeper et al., 2015). The protein sequences were predicted using MEGAX (Kumar et al., 2018).

Mutations were investigated in all genes of animal isolates and compared to human isolates. Only differences between animal or animal-human isolates were reported. The frequency of a mutation was calculated as follows: Frequencyofamutation=ThenumberofgenomesthathavethemutationThetotalnumberofgenomes×100

All mutations with their frequency and effects on the amino acid sequences are reported in Table S2. Mutations that existed in more than 2% of the studied genomes for a given animal species are reported in the main tables (Tables 1–3). Variations that existed only between human isolates were not reported. The effect of substitution mutations on the amino acid sequence was determined. Some viral genomes such as isolates from the cat, dog, mink, and human were incomplete (having long stretches of Ns, see 3 Table S1 for more information on the percentage of Ns in each of the studied genomes). These ambiguous genomic regions were not used for mutation analysis for those species.

Table 1 Amino acid substitutions in the non-structural proteins encoded by the orf1ab gene of SARS-CoV-2 isolated from animals compared to humans.

GeneA	Amino acid position	SARS-CoV-2 Reference	Cat	Dog	Mink	Mouse	Tiger	Human	
nsp2	265	T	I (20.0)	T	T	T	I (100.0)	T or I	
352	E	E	E	Q (2.7)	E	E	E	
372	A	A	A	V (46.3)	A	A	A	
388	H	Y (40.0)	H	H	H	H	H	
398	R	R	R	C (25.2)	R	R	R	
405	A	A	A	T (4.8)	A	A	A	
743	E	E	E	V (2.7)	E	E	E	
nsp3	953	D	Y (20.0)	D	D	D	D	D	
996	D	D	D	Y (20.7)	D	D	D or Y	
1052	V	I (40.0)	V	V	V	V	V	
1096	P	P	P	L (3.4)	P	P	P	
1113	H	H	H	Y (4.1)	H	H	H	
1202	K	N (40.0)	K	K	K	K	K	
1568	I	I	I	V (10.9)	I	I	I	
1588	M	M	M	K (20.0)	M	M	M	
2101	D	D	G (50.0)	D	D	D	D	
nsp4	3076	H	H	H	H (100.0)	Y	H	H or Y	
3233	H	H	Y (50.0)	H	H	H	H or Y	
nsp5	3512	I	T (20.0)	I	I	I	I	I	
3522	I	I	I	V (5.4)	I	I	I	
nsp6	3606	L	L	L	F (12.2)	L	L	L or F	
3615	A	A	A	V (25.9)	A	A	A or V	
nsp9	4177	G	G	G	E (28.6)
R (23.8)	G	G	G	
nsp10	4377	K	E (16.7)	K	K	K	K	K	
nsp12	4418	T	I (40.0)	T	T	T	T	T	
4588	M	M	M	I (2.7)	M	M	M	
4715	P	L (100.0)	L (50.0)	L (74.8)	P	L	P or L	
5195	T	T	T	I (4.1)	T	T	T	
nsp13	5582	I	I	I	V (6.1)	I	I	I	
5716	R	R	R	C (25.2)	R	R	R or C	
5770	A	A	A	D (3.0)	A	A	A	
nsp15	6544	A	A	A	T (2.8)	A	A	A	
Notes:

A Orf1ab, open reading frame 1ab; nsp, non-structural protein.

The numbers in brackets represent the percentage of genomes that contain the substitution. For example, the substitution of amino acid number 372 of nsp2 protein from A to V (A372V) in mink occurred in 46.3% of mink isolates. Percentages are reported for substitutions only. Substitutions that existed in less than 2% of the genomes are not reported. Unique mutations for an animal species are highlighted in yellow.

Table 2 Amino acid substitutions in the structural proteins of SARS-CoV-2 isolated from animals compared to humans.

Gene	Amino acid position	SARS-CoV-2 Reference	Cat	Dog	Mink	Mouse	Tiger	Human	
SA	8	L	L	V (50.0)	L	L	L	L or V	
261	G	G	G	D (4.8)	G	G	G	
262	A	A	A	S (5.4)	A	A	A	
367	V	V	V	F (5.4)	V	V	V	
452	L	L	L	M (10.2)	L	L	L	
453	Y	Y	Y	F (23.8)	Y	Y	Y	
486	F	F	F	L (17.0)	F	F	F	
498	Q	Q	Q	Q	H (100.0)	Q	Q	
501	N	N	N	T (3.4)	N	N	N or Y	
614	D	G (100.0)	D	G (74.8)	D	G	D or G	
969	N	N	N	N	S (100.0)	N	N	
MB	3	D	D	D	G (2.7)	D	D	D or G	
175	T	M (16.7)	T	T	T	T	T	
NC	41	R	R	R	L (2.7)	R	R	R	
80	P	P	P	L (2.7)	P	P	P	
199	P	P	P	Q (2.7)	P	P	P	
203	R	K (60.0)	R	R	R	R	R or K	
204	G	R (60.0)	G	G	G	G	G or R	
247	T	I (16.7)	T	T	T	T	T	
Notes:

A S, spike.

B M, matrix.

C N, nucleocapsid.

The numbers in brackets represent the percentage of genomes that contain the substitution. For example, the substitution of amino acid number 261 of S protein from G to D (G261D) in mink occurred in 4.8% of mink isolates. Percentages are reported for substitutions only. Substitutions that existed in less than 2% of the genomes are not reported. Unique mutations for an animal species are highlighted in yellow. No substitutions were detected in the envelope protein.

Table 3 Amino acid substitutions in the non-structural proteins encoded by orf3a and orf6 genes of SARS-CoV-2 isolated from animals compared to humans.

Gene	Amino acid position	SARS-CoV-2 Reference	Cat	Dog	Mink	Mouse	Tiger	Human	
orf3aA	57	Q	Q	Q	H (25.9)	Q	H	Q or H	
182	H	H	H	L (3.4)
Y (22.4)	H	H	H	
219	L	L	L	V (25.6)	L	L	L	
224	G	G	G	C (2.7)	G	G	G	
229	T	T	T	I (16.3)	T	T	T	
251	G	G	V (50.0)	G	G	G	G or V	
orf6B	20	R	R	R	S (10.2)	R	R	R	
23	K	K	K	S (2.0)	K	K	K	
Notes:

A orf3a, open reading frame 3a.

B ns3a, Non-structural protein 3a.

The numbers in brackets represent the percentage of genomes that contain the substitution. For example, the substitution of amino acid number 219 of ns3a protein (encoded by orf3a) from L to V (L219V) in mink occurred in 25.6% of mink isolates. Percentages are reported for substitutions only. Substitutions that existed in less than 2% of the genomes are not reported. Unique mutations for an animal species are highlighted in yellow.

Evolutionary divergence

The pairwise genetic distances (Table S3) were calculated in MEGAX using the Maximum Composite Likelihood model (Tamura, Nei & Kumar, 2004). The genetic distances were estimated for forty-two sequences representing SARS-CoV-2 isolates from humans and animals, pangolin coronavirus (MT121216.1), and bat coronavirus RaTG13 (MN996532.2). The sequence identity percentage was calculated from the pairwise genetic distance as follows: Identity%=100−(pairwisedistance×100)

Phylogenetic analysis

Forty SARS-CoV-2 genomes were used for phylogenetic analysis (Fig. 1). The phylogenetic tree was constructed based on the coding sequences of all genes. The 3′ UTR, 5′ UTR, and non-coding sequences were excluded. Sequences were aligned using Kalign3 (Lassmann, 2020). The phylogenetic tree was constructed in IQ-TREE (Nguyen et al., 2015) using the maximum likelihood method, ModelFinder (to identify the best fitting model (s)), and ultrafast bootstrap approximation (1,000 replicates). The best-fitting models were TN (Tamura-Nei) + F (Felsenstein) + I (proportion of invisible sites). The tree was drawn to scale with branch length representing the number of substitutions per site. The tree was visualized in MEGAX (Kumar et al., 2018) and rooted on the midpoint.

Figure 1 Phylogenetic tree of 40 SARS-CoV-2 genomes from animals and humans.

Each sequence is reported by the accession number, host, and geographic region (country). Different clades are labeled and highlighted differently. The tree was constructed in IQ-TREE using the maximum likelihood method, ModelFinder, and ultrafast bootstrap approximation (1,000 replicates). The tree is rooted on midpoint. Branch lengths are measured in the number of substitutions per site. The percentage of replicate trees in which the associated viruses clustered together in the bootstrap test is represented by the numbers above the branches. Bootstrap values less than 50% are not shown.

Results

Sequence analysis of SARS-CoV-2 isolated from animals

The SARS-CoV-2 isolated from animals had the same genome organization and encoded the same proteins as human isolates. The nucleotide sequence identities of the whole genome of SARS-CoV-2 animal isolates were compared with different SARS-CoV-2 human isolates (Table S3). The homology analyses of human coronaviruses with hCoV-19/tiger, hCoV-19/mouse/Harbin, hCoV-19/mink, hCoV-19/cat, hCoV-19/canine (dog, EPI_ISL_450403) showed a percent identity of 99.7–100%, 99.95–99.98%, 99.94–99.96%, 99.97–99.98%, and 99.97–99.99% with other human SARS-CoV-2 isolates, respectively. The phylogenetic tree was reconstructed based on the coding sequences of the whole genome (Fig. 1). Most animal and human SARS-CoV-2 isolates from the same GISAID clade clustered together. For example, the clade GR cluster contained the sequences from the dog, cats, and humans (all belong to the clade GR) regardless of their geographic region. Similar results were obtained for isolates from the mink, tiger, and mouse. In the case of clade G, human and animal isolates from the Netherlands clustered separately from other countries (Fig. 1).

Mutational spectra of non-structural proteins (ORF1ab)

Sequence analysis of the ORF1ab gene of the studied isolates from animals revealed a total of 133 substitutions, 108 of which were unique in animals (Table S2). Most of the unique mutations (98) occurred in mink isolates, while 2, 6, and 2 mutations were reported in dog, cat, and mouse isolates, respectively. Of the 133 mutations, 79 were non-synonymous (changed amino acid sequence), while 54 were synonymous.

To summarize the results and exclude less frequent mutations, we set a threshold of 2% for reporting mutations in Tables 1–3. Table 1 contains the most common amino acid substitutions (above the threshold) of the non-structural proteins encoded by the ORF1ab gene of SARS-CoV-2. The nsp2 exhibited six unique amino acid substitutions, one in cat isolate (H388Y) and five in mink isolates (E352Q, A372V, R398C, A405T, and E743V), while nsp3 (Papain-like proteinase domain) revealed three (D953Y, V1052I, and K1202N), one (D2101G), and four (P1096L, H1113Y, I1508V, and M1588K) unique amino acid substitutions in SARS-CoV-2 isolates from the cat, dog, and mink, respectively. The nsp5 (3C-like proteinase domain) had two unique amino acid substitutions, one in cat isolate (I3512T) and the other in mink isolate (I3522V). Moreover, nsp9 (RNA/DNA binding activity) and nsp15 (Poly(U) specific endoribonuclease) of SARS-CoV-2 isolated from the mink exhibited one unique amino acid substitution at (G4177E or R) and (A6544T), respectively. In addition, nsp10 of SARS-CoV-2 isolated from the cat displayed one unique amino acid substitution (K4377E), while nsp12 (RNA-dependent RNA polymerase, RdRp) exhibited three unique amino acid substitutions, one for the cat (T4418I) and two for the mink (M4588I and T5195I). Two unique amino acid substitutions (I5582V and A5770D) were recorded in mink isolate for nsp13 (helicase). Finally, the frequency of mutations in nsp1, nsp4, nsp6, nsp7, nsp8, nsp11, nsp14, and nsp16 of the ORF1ab gene was less than the 2% threshold, therefore not listed in Table 1.

Mutational spectra of structural proteins

Sequence analysis of SARS-CoV-2 structural genes from animals revealed several mutations in S, M, and N gene, while no mutations were detected in the E gene.

Mutational spectra of spike (S) protein

The spike gene of animals SARS-CoV-2 consists of 3,822 nucleotides (21,563–25,384) that encode 1,273 amino acids. The domain structures and critical ACE2-binding residues of S protein from animal SARS-CoV-2 are illustrated in Fig. 2. The analysis of mutations in the S gene of the studied isolates from animals revealed a total of 25 unique substitution mutations, 22 of which occurred in mink isolates (Table S2). The cat isolates had one unique substitution, while the mouse had two unique substitutions and one deletion mutation. No mutations were identified in the S gene from the dog or tiger isolates. Of the 25 substitutions, 17 were non-synonymous, while nine substitutions were synonymous (Table S2).

Figure 2 Map of SARS-CoV-2 spike protein structure.

This includes N-terminal end (NH2) to the C-terminal end (COOH), the signal peptide (SP), N-terminal domain (NTD), receptor-binding domain (RBD), receptor-binding motif (RBM), fusion peptide (FP), heptad repeat 1 (HR1), heptad repeat 2 (HR2), transmembrane domain (TM), and intracellular domain (IC). The spike protein consists of two subunits: S1 and S2 that are cleaved at the furin cleavage site. The differences in the spike amino acid sequences from animal isolates compared to SARS-CoV-2 reference (Wuhan-Hu-1 isolate), pangolin coronavirus MP789, and bat coronavirus RaTG13 are marked by the red rectangles. The multiple sequence alignment is presented for the RBM and the furin cleavage site.

The non-synonymous mutations reported for spike protein in Table 2 were distributed as follows: one in the signal peptide (SP), two in the N-terminal domain (NTD), six in the receptor-binding domain (RBD), one downstream of the RBD (D614G), and one in the heptad repeat 1 (HR1) (Refer to Fig. 2 for the map of spike protein). In the mouse isolate, two unique amino acid substitutions (Q498H and N969S) were identified, with one of them (Q498H) located within the RBM (Fig. 3). Fifteen unique mutations were identified in mink isolates (Table S2), four of which (L452M, Y453F, F486L, N501T) were located within the RBM of S protein (Figs. 2 and 3; Table 2). The deletion of five amino acid residues in the isolate from the mouse was close to the furin cleavage site. No mutations were identified at the furin cleavage site of SARS-CoV-2 isolates from other animals. All mutations reported for spike protein in Table 2 are unique for animal isolates except the mutation in the SP for the dog (L8V). The position and structure of each amino acid substitution in the RBM compared to normal residues are illustrated in Fig. 3.

Figure 3 Amino acid substitutions in the receptor-binding motif (RBM) of SARS-CoV-2 S protein from animal isolates.

Only monomeric partial structure of S protein is shown where the receptor-binding domain (RBD) is purple, RBM is blue, and the substitution sites are red. This structure is based on Protein Data Bank (PDB) reference structure 6ZB4. In each mutation site, the side chains of the amino acid residues are shown for normal (A) and mutated (B) RBM.

Mutational spectra of matrix (M) protein

The M gene of SARS-CoV-2 isolated from animals is composed of 669 nucleotides (26,523–27,191) that encode 222 amino acids. The M protein consists of the N-terminal domain that is present on the virion surface, and the C-terminal domain that is located on the interior surface of the virion. Analysis of mutations in the M gene of the studied animal isolates revealed a total of six unique nucleotide substitution mutations, five of which occurred in mink isolates and one in the cat isolate (Table S2). Four of the six mutations were non-synonymous, while two were synonymous. No mutations were recorded in the M gene of the dog, mouse, or tiger (Table S2). The frequency of the non-synonymous mutations was below the 2% threshold, therefore not reported in Table 2 except the T175M substitution in the cat isolate (16.7%).

Mutational spectra of nucleocapsid (N) protein

The N gene of SARS-CoV-2 isolated from animals is composed of 1,260 nucleotides (28,274–29,533) that encode 419 amino acids. Analysis of N protein sequences from animal SARS-CoV-2 isolates revealed 19 unique mutations, 17 of which occurred in mink and two in cat (Table S2). No mutations were detected in the N gene of the dog, mouse, or tiger. Twelve of the 19 mutations were non-synonymous (Table S2). SARS-CoV-2 isolated from the mink exhibited three unique amino acid substitutions (R41L, P80L, and P199Q) with a frequency of 2.7%, while SARS-CoV-2 isolated from the cat had one unique amino acid substitution (T247I) with a frequency of 16.7% (Table 2).

Mutational spectra of accessory proteins

Severe acute respiratory syndrome coronavirus-2 genome has six accessory genes that encode six accessory proteins: ORF3a (275 aa), ORF6 (61 aa), ORF7a (121 aa), ORF7b (43 aa), ORF8 (121 aa), and ORF10 (38 aa). No mutations were identified in ORF10, while sixteen, three, two, one, and three nucleotide substitutions were discovered in ORF3a, ORF6, ORF7a, ORF7b, and ORF8 of mink isolates, respectively (Table S2). In addition, 134-nucleotide deletion mutation, resulting in the deletion of 45 amino acid residues, was identified in ORF7a of two mink isolates (EPI_ISL_447623 and EPI_ISL_522991). Four unique amino acid substitution positions (H182L or H182Y, L219V, G224C, T229I) within the ORF3a protein of mink isolates had a frequency of more than 2%. Also, two unique substitutions (R20S and K23S) in the ORF6 protein of the mink isolates had a frequency of more than 2% (Table 3).

Discussion

The novel coronavirus (SARS-CoV-2) has been identified as the causative agent of the coronavirus disease 2019 (COVID-19) outbreak in Wuhan City, Hubei province, China in December 2019 (Huang et al., 2020; Zhou et al., 2020). Like other coronaviruses, the interspecies transmission of SARS-CoV-2 is possible. Bats are the natural host of human coronaviruses 229E, SARS-CoV, NL63, and MERS-CoV (Corman et al., 2015, 2016; Donaldson et al., 2010; Ge et al., 2013; Hu et al., 2015; Huynh et al., 2012; Samara & Abdoun, 2014; Tao et al., 2017), while rodents are the natural host of HKU1 and OC43 (Cui, Li & Shi, 2019). The intermediate hosts of both NL63 and HKU1 are not yet unidentified, while the intermediate hosts of 229E, OC43, SARS-CoV, and MERS-CoV are camelids, bovines, palm civets, and dromedary camels, respectively (Corman et al., 2015; Cui, Li & Shi, 2019; Donaldson et al., 2010; Ge et al., 2013; Hu et al., 2015). This study aimed to identify the genomic features that might be associated with interspecies jumping of SARS-CoV-2 between humans and animals. To this extent, SARS-CoV-2 genomes from different species including humans, cats, dogs, mouse, minks, and tiger were fully characterized and compared with geographically and phylogenetically close SARS-CoV-2 human isolates.

Phylogenetic analyses of hCoV-19/tiger, hCoV-19/mouse/Harbin, hCoV-19/mink, hCoV-19/cat, and hCoV-19/dog revealed a close relationship (99.7–100%) with SARS-CoV-2 human isolates. It is speculated that bats are the natural source of SARS-CoV-2, hence the phylogenetic analysis of SARS-CoV-2 revealed a nucleotide homology of 96.3% with the bat CoV RaTG13 (Zhou et al., 2020). Additionally, the nucleotide sequence homology between pangolin and analyzed human SARS-CoV-2 isolates in this study is 89%. Although pangolin could not be excluded as one of the intermediate animal hosts of SARS-CoV-2 (Lam et al., 2020), a direct transmission of pangolin SARS-CoV-2 might not be possible due to the sequence divergence.

Generally, coronaviruses can easily cross species barriers. There are three main reasons for interspecies transmission of coronaviruses; firstly, the high mutation rates (Su et al., 2016), although the mutation rate of SARS-CoV-2 is lower than SARS-CoV (Ye et al., 2020). Secondly, the large RNA genome of coronaviruses, which increases the incidence of mutations and recombination leading to the emergence of novel CoVs (Ye et al., 2020). Thirdly, the interaction of coronaviruses with different ACE2 receptors (Bolles, Donaldson & Baric, 2011). The SARS-CoV and HCoV-NL63 bind with ACE2 for interspecies transmission (Graham & Baric, 2010; Hofmann et al., 2005). However, HCoV-229E and HCoV-OC43 use aminopeptidase N (APN) and 9-O-acetylated sialic acid (9-O-Ac-Sia) receptors, respectively, which helps successful adaptation in humans after interspecies transmission from their animal hosts (Huang et al., 2015; Liu, Liang & Fung, 2020). Although SARS-CoV-2 uses the ACE2 receptor for host cell entry (Hoffmann et al., 2020; Zhou et al., 2020), it remains unclear whether any other coreceptor might be required for its transmission.

In the present study, sequence analysis of the SARS-CoV-2 revealed several unique amino acid substitutions within the S, M, N, and nsps proteins. These mutations may indicate that the virus is not yet adapted to these animals. Duffy (2018) reported that the mutation rate is often high when coronaviruses are not well adapted to their hosts. Several amino acid substitutions may not significantly switch the amino acid properties as the involved amino acid residues were either from the same class or shared some properties. In S protein, for example, L452M involved leucine and methionine, and both are neutral and hydrophobic. Similarly, Y453F involved tyrosine and phenylalanine, and both are hydrophobic and aromatic. However, Q498H in mouse isolate may have a significant change in the properties of the amino acid residues because glutamine is neutral while histidine is positively charged, and both have significant differences in the structure of the side chain.

Mutations in the S protein could change the tropism of the virus, pathogenicity, and interspecies transmission (Wan et al., 2020). In our study, several unique mutations were found in the SP, NTD, RBD, downstream to RBD, and in the HR1 of S protein. Ortega et al. (2020) reported that slight modifications of some residues within RBD of SARS-CoV-2 could improve the interaction with the human cellular receptors. The D614G mutation downstream of the RBD could be associated with higher transmission, pathogenicity, and evasion of immune interventions (Happi et al., 2020; Korber et al., 2020). It was found that K479N and S487T of SARS-CoV are associated with human ACE2 receptor recognition. These amino acids are corresponding to Q493 and N501 in SARS-CoV-2, respectively (Ortega et al., 2020). In our study, all analyzed SARS-CoV-2 from animals had Q493 and N501 except mink isolates which had N501T (Figs. 2 and 3). However, the question whether other reported mutations contribute to further adaptation, and subsequently impact the virus pathogenicity require further investigation. Computational models and data have identified additional mutations that might further strengthen the binding affinity of SARS-CoV-2 to ACE2 receptors (Wan et al., 2020). Yao et al. (2020) found that S protein mutations of SARS-CoV-2 are capable of changing its pathogenicity. Further studies are required to investigate the impact of these mutations on virus evolution.

The N protein of SARS-CoV-2 consists of 1,260 nucleotides that encode 419 amino acids. It is a multifunction protein and an important determinant of the virus pathogenicity, viral transcription efficiency, and coiling the RNA segment into a helical nucleocapsid incorporated into virion of icosahedral symmetry (McBride, Van Zyl & Fielding, 2014; Paraskevis et al., 2020). In our study, the N protein had four amino acid substitutions: three in mink isolates (R41L, P80L, and P199Q) with a frequency of 2.7% each, and one in cat isolates (T247I) with a frequency of 16.7%.

Generally, the nsps (nsp1-16) of coronaviruses play essential roles in virus replication (Lei, Kusov & Hilgenfeld, 2018; Zeng et al., 2018), polypeptide cleaving (Lei, Kusov & Hilgenfeld, 2018; Zhu et al., 2017), and inhibition of host immune response (Lei, Kusov & Hilgenfeld, 2018; Shi et al., 2019; Zhu et al., 2017). The ORF1ab polyprotein of SARS-CoV-2 plays an important role in viral RNA synthesis (Banerjee et al., 2020). In SARS CoV, it was found that nsp2 binds to the host proteins (prohibitin 1 and prohibitin 2) which are involved in several cellular functions including cell cycle progression, cell migration, cellular differentiation, apoptosis, and mitochondrial biogenesis (Cornillez-Ty et al., 2009). Analysis of ORF1ab nsp2 (638 aa from A181-G818) polyproteins of SARS-CoV-2 isolated from animals revealed six characteristic mutations, one in cat (H388Y) and five in mink (E352Q, A372V, R398C, A405T, and E743V) isolates. The impact of these mutations on the function of nsp2 and virus evolution is still unknown, although stabilizing mutation falling in the endosome-associated-protein-like domain of the nsp2 protein, could explain why SARS-CoV-2 is more contagious than SARS-CoV (Angeletti et al., 2020).

The papain-like proteinase (PLpro) and 3C-like main protease (3CLpro) are encoded by nsp3 and nsp5, respectively. Both PLpro and 3CLpro play a role in virus replication and antagonize the host’s innate immunity (Harcourt et al., 2004; Li et al., 2016). They are also a popular target for antiviral drugs (Wu et al., 2020). Sequence analysis of the nsp3 and nsp5 of SARS-CoV-2 isolated from animals revealed eight and three unique amino acid substitutions, respectively. The role of these mutations in nsp3 on virus replication and susceptibility to antiviral drugs should be investigated. The nsp3 mutation could explain the difference in pathogenicity between SARS-CoV-2 and SARS-CoV (Angeletti et al., 2020). Furthermore, unique mutations were found within the nsp9, nsp10, nsp12, and nsp13. Although no direct evidence exists about their roles, continued evidence-based analysis of evolutionary change is important.

Additionally, the ORF3a protein of coronaviruses showed a pro-apoptotic activity. The SARS-CoV-2 is less virulent than SARS-CoV due to diminished pro-apoptotic activity (Ren et al., 2020). Herein, different mutations were found within ORF3a (Table 3; Table S2). However, the impact of mutations within ORF3a of SARS-CoV-2 on the apoptotic activity and consequently on the virus pathogenicity is unknown.

In conclusion, although the dynamics of SARS-CoV-2 transmission from humans to animals and vice versa is still unclear, further human-animals interface is possible due to the similarity of ACE2 receptors in humans and some animal species. In the present study, unique amino acid substitutions were reported in almost all animal SARS-CoV-2 proteins such as the ORF1ab, S glycoprotein (including critical ACE2-binding sites), ORF3a, and N proteins. Our study draws the attention of scientists worldwide on the role of mink as an animal host to SARS-CoV-2 due to several unique amino acid substitutions recorded in its proteins (specially S protein) that might play a role in the virus evolution, pathogenesis and transmission to human. This study also recommends continuous molecular surveillance of SARS-CoV-2 animal isolates. Also, it is essential to furtherly investigate if these mutations could impact the virus evolution, including virus adaptation, changes in the pathogenicity, and the emergence of drug-resistance viral phenotypes.

Supplemental Information

Supplemental Information 1 Information on coronavirus genomes used in the current study.

Click here for additional data file.

Supplemental Information 2 The nucleotide substitutions identified in SARS-CoV-2 genomes from five animals (cat, dog, mink, mouse, and tiger), their position within the gene and the genome, the codon change, the effect on the amino acid, and the mutation frequency.

Click here for additional data file.

Supplemental Information 3 Pairwise genetic identity percent for SARS-CoV-2 isolates from humans, animals, reference, bat coronavirus RaTG13, and pangolin coronavirus (MT121216.1).

Click here for additional data file.

We gratefully acknowledge the authors, originating, and submitting laboratories of the sequences from the GISAID’s EpiCoV™ Database and NCBI on which this study is based.

Additional Information and Declarations

Competing Interests

Author Contributions

Data Availability

Awad A. Shehata is employed at PerNaturam GmbH. This company is interested in the production of natural products and have no activities related to microbiology.

Ahmed Elaswad performed the experiments, analyzed the data, prepared figures and/or tables, authored and reviewed drafts of the paper, and approved the final draft.

Mohamed Fawzy performed the experiments, analyzed the data, authored and reviewed drafts of the paper, and approved the final draft.

Shereen Basiouni performed the experiments, analyzed the data, and approved the final draft.

Awad A. Shehata conceived and designed the experiments, performed the experiments, analyzed the data, authored and reviewed drafts of the paper, and approved the final draft.

The following information was supplied regarding data availability:

Raw data, including GenBank and GISAID accession numbers, are available in the Supplemental Files.

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
