# Peer review of "Mutational spectra of SARS-CoV-2 isolated from animals"

_PeerJ, doi:10.7717/peerj.10609_

## Round 0.1 · original submission · Major Revisions

Dear Dr. Elsawad and colleagues:

Thanks for submitting your manuscript to PeerJ. I have now received three independent reviews of your work, and as you will see, the reviewers raised some major concerns about the research. Despite this, these reviewers are optimistic about your work and the potential impact it will have on research studying the evolution and epidemiology of SARS-COV-2. Thus, I encourage you to revise your manuscript, accordingly, taking into account all of the concerns raised by the reviewers.

There are major concerns regarding your sampling, both in terms of the limited sample size relative to available SARS-COV-2 sequences, as well as the use of outgroup SARS sequences. There are several great suggestions by the reviewers that I would like to see followed.

Importantly, please ensure that an English expert has edited your revised manuscript for content and clarity. Please also ensure that your figures and tables contain all of the information that is necessary to support your findings and observations.

There are many comments by the reviewers that ask for more information on specific issues; please address these.

I look forward to seeing your revision, and thanks again for submitting your work to PeerJ.

Good luck with your revision,

-joe

·

Basic reporting

The manuscript “Mutational spectra of SARS-COV-2 isolated from animals” is summarized the nucleotide and amino acid substitutions at different gene segments and proteins of SARS-COV-2 isolates. Authors tried to explain the comparison of the mutational effect of SARS-COV-2 in different animal species and human. However, the objective of the study and the effect of mutation should better explain which would benefit the reader in a greater extent. Similarly, the tables are full of content and difficult to understand. I suggest to revise the Tables and prioritize them in main tables and supplemental tables. I also recommend to use simple and fluent English language to easily understand the text.

There are some specific major issues to be addressed:
1. Please consider doing a minor check across the manuscript for English usage, syntax and elaborate the abbreviations when comes first. For example: in line 72; the full name “pp”1ab is missing, in line 110; NCBI and GISAID, in line 258, 287; space is missing in between “animalisolate”.
2. In line 100-101: the sentence (SARS COV-2 can infect …………………………chicken and duck) is a bit confusing. Make the statement clear with appropriate references. For eg., according to Shi et al., SARS COV 2 have poorly replicate in chicken and duck and also compare the statement with others observation eg. Schlottau et al., 2020
(https://www.thelancet.com/journals/lanmic/article/PIIS2666-5247(20)30089-6/fulltext)
3. In the line 102: before mentioning the objectives, I suggest you to explain briefly the justification of the study.
4. In line 184-186 (All SARS-CoV-2……………….at this position): not clear either animal virus substitutions are different from the human virus isolate. In the Table and Figure the substitutions are similar in animal viruses and human viruses.
5. Line 233-235: rephrase the sentence to better understand.
6. Line 287-290: Mentioned signature mutation for which species? What is the relation of this statement with the study?
7. Line 298-299: The author stated animal have a role in the SARS-COV-2 outbreak but is not these animal viruses isolated after the pandemic? Whether the transmission is not the other way around? Please give enough statement and references to support the conclusions.
8. Please explain the Figure 2 legend to understand better. The first paragraph of the legend is not completed (heptad rep….).
9. Synchronization of the title and content of the Table 2 is missing. Proteolytic cleavage site is missing on the table 2. Please correct either the content or edit the title accordingly.
10. Table 3 is unclear and difficult to understand. Please explain each headline and mention the effect of amino acid substitution (either bad or good) with appropriate references. What is the yellow mark stand for? If necessary, please use footnote to explain better. Similar observation for table 4.

Experimental design

Research questions are not well defined. Need further improvement.

Methods are well defined.

Validity of the findings

Results are explained well but discussion need to be specific and well written.

Speculation and conclusions need more supportive statement.

Reviewer 2 ·

Basic reporting

The current manuscript Ahmed Elsawad et al has tried to identify mutations within SARS-CoV-2 strains isolated from different animals. The idea is really good and worth exploring.

The introduction part is majorly written using content from Helmy et al., 2020. There are several places in the introduction which are approximately similar to Helmy et al., 2020. There are numerous publications available on the SARS-CoV-2 genome and COVID-19 disease. The introduction of any article is like a mini-review of our current knowledge procured from diverse publications, therefore my suggestion is to re-write the introduction section using that diverse information.

At several places, sentences should be rephrased such as line 65 should be "still the disease is spreading worldwide".

Experimental design

Considering the vastness of SARS-CoV-2 genome availability, it becomes tricky to select genome sequences. I do not understand on the basis of which criteria authors have selected those 38 human SARS-CoV-2. There are multiple clades diversifying the SARS-CoV-2 genomes based on diverse mutation sites, authors have not discussed any point in this regard.

Line 136-138: The authors should mention how many Ns are present in Cat and Dog CoV.

Validity of the findings

The phylogeny shown in Figure 1 is really vague and does not depict any real finding. As SARS-CoV-2 genomes from other animals (leaving bat and pangolin) are >99% identical, the SARS-CoV-2 clade is not showing any relationship amongst these strains. As the addition of bat CoV RaTG13, pangolin CoV and other CoVs will make the relative distance between SARS-CoV-2 genomes too high, It would be better to use only SARS-CoV-2 genomes and see how they are related with each other.

Wuhan-Hu-1 is the reference strain, however, now it is already known that there are more than 10 clades within SARS-CoV-2. The authors should explain the clade of each used strain. Later, they should use different representatives of diverse SARS-CoV-2 clades and classify these animal strains amongst any of the known clade or a diverse clade based on phylogeny and alignments.

Line 223-225: This is wrong speculation "This supports the speculation that animals have a role in the evolutionary pathway of SARS-CoV-2." It does not prove anything like this. These animals are just getting infected with the Human CoV (as they are >99.5% identical) and these are not in the line of SARS-CoV-2 common ancestors as in Pangolin and Bat-CoVs.

Authors should also explain synonymous or nonsynonymous mutations in each case. Simultaneously, They should explain that the identified mutation spectra belong to a transition or transversion event?

Line152-164: are these lines really required? You can combine this information in Table 2.

Line 258: B is missing from Bat.

The authors should provide more information in Figure 2. They should try to show the alignment for all descriptions in Line 178-189.

Table 2 is more like a Review figure.

Table 3: it is not correct to say alleles

Reviewer 3 ·

Basic reporting

THe overall representation of the manuscript is good and acceptable.

Experimental design

The bioinformatics analysis can be more rigorous.

Validity of the findings

The findings are genotypic and its impact on the structure of the proteins needs multiple analysis.

Additional comments

IN the manuscript entitled "Mutational spectra of SARS-COV-2 isolated from
animals", the authors tried to correlate the evolutionary pattern of SARS-CoV-2 in animals. Unfortunately, the data set is very less (n=55) and out of which only 7 genomes for
are from animals (cat, dog, mink, mouse, and tiger) and the entire manuscript deals with the mutations around these animals.
I think the dataset should be increased, so that mutations pattern can be assured. Simultaneously, the authors should analyse the impact of these mutations.

---

## Round 0.2 · accepted · Accept

Dear Dr. Elaswad and colleagues:

Thanks for revising your manuscript based on the concerns raised by the reviewers. I now believe that your manuscript is suitable for publication. Congratulations! I look forward to seeing this work in print, and I anticipate it being an important resource for groups studying the evolution and epidemiology of SARS-COV-2. Thanks again for choosing PeerJ to publish such important work.

Best,

-joe

·

Basic reporting

The revised manuscript “Mutational spectra of SARS-COV-2 isolated from animals” is summarized and compared the nucleotide and amino acid substitutions at different gene segments and proteins of SARS-COV-2 isolates from animals and human.” The manuscript has much improved and addressed most of the comments that has been previously raised.

Experimental design

Aims and objectives are clear in the revised version

Validity of the findings

The findings specially tables are much improved and understandable.
Data are well defined and controlled.

Additional comments

The author addresses most of the previous comments and improved the text and sequence analysis in a sound and scientific manner.